# The protein translation machinery is expressed for maximal efficiency in *Escherichia coli*

Xiao-Pan Hu [1], Hugo Dourado [1], Peter Schubert [1] & Martin J. Lercher [1✉]

Protein synthesis is the most expensive process in fast-growing bacteria. Experimentally observed growth rate dependencies of the translation machinery form the basis of powerful phenomenological growth laws; however, a quantitative theory on the basis of biochemical and biophysical constraints is lacking. Here, we show that the growth rate-dependence of the concentrations of ribosomes, tRNAs, mRNA, and elongation factors observed in *Escherichia coli* can be predicted accurately from a minimization of cellular costs in a mechanistic model of protein translation. The model is constrained only by the physicochemical properties of the molecules and has no adjustable parameters. The costs of individual components (made of protein and RNA parts) can be approximated through molecular masses, which correlate strongly with alternative cost measures such as the molecules' carbon content or the requirement of energy or enzymes for their biosynthesis. Analogous cost minimization approaches may facilitate similar quantitative insights also for other cellular subsystems.

[1] Institute for Computer Science and Department of Biology, Heinrich Heine University, D-40225 Düsseldorf, Germany. ✉email: martin.lercher@hhu.de

Protein translation is central to the self-replication of biological cells. While the workings of its individual components are well understood, the translation apparatus is a complex machine with many degrees of freedom, where the same rate of protein production could be achieved with very different relative abundances of its components. Although a large body of quantitative experimental data on these abundances in *E. coli* across different growth conditions is available, it is still unclear according to which organizing principle(s)—if any—they are set by the cell. Given that translation is the energetically most expensive process in fast growing *E. coli* cells, accounting for up to 50% of the proteome[1] and 2/3 of cellular ATP consumption[2], it is likely that natural selection acted to optimize the efficiency of translation. But what exactly is efficiency in the evolutionary context?

In the late 1950s, it was hypothesized that ribosomes operate at a constant, maximal rate[3,4], consistent with the observed linear dependence of ribosome concentration on growth rate[3,5–7]. This hypothesis was later proven untenable, as the activity of ribosomes was observed to increase with growth rate[8]. Klumpp et al.[9] suggested that optimal translational efficiency corresponds to the parsimonious usage of translation-associated proteins, most notably ribosomal proteins, elongation factor Tu, and tRNA synthetases. While these authors were able to fit a coarse-grained phenomenological model to the data, their suggested evolutionary objective could also not explain the observed growth rate dependencies quantitatively (see Supplementary Note 1 for a discussion of ref. [9] and of the phenomenological model of bacterial growth it is based on[5,10]). Thus, it is currently unclear to what extent translation has indeed been optimized by natural selection, and—if such optimization indeed occurred—whether its action can be expressed in terms of a simple objective function.

In principle, these questions would best be addressed in the context of a whole-cell model of balanced growth that combines mechanistic descriptions of metabolism and protein production. However, while such models have been described conceptually[11,12], kinetic parameterizations are unavailable for a majority of the relevant enzymatic reactions[13], preventing a truly mechanistic description that combines metabolism and protein translation. Thus, we here focus on protein production alone, taking the experimentally observed output of translation (proteome production rate and composition in a given growth condition), the corresponding input (charged tRNAs), and the kinetics of individual translation reactions as given. We then use this mechanistic description of translation to find the combination of the concentrations of mRNA, ribosomes, elongation factors, and tRNAs that results in minimal cellular costs in the given condition. Thus, our estimate of the optimal efficiency of the translation machinery is not based on the maximization of ribosome activity, but on the minimization of the combined cost of the complete translation machinery at an observed protein production output.

We base our cost definition on the experimental observation that cellular dry mass per cell volume is approximately constant

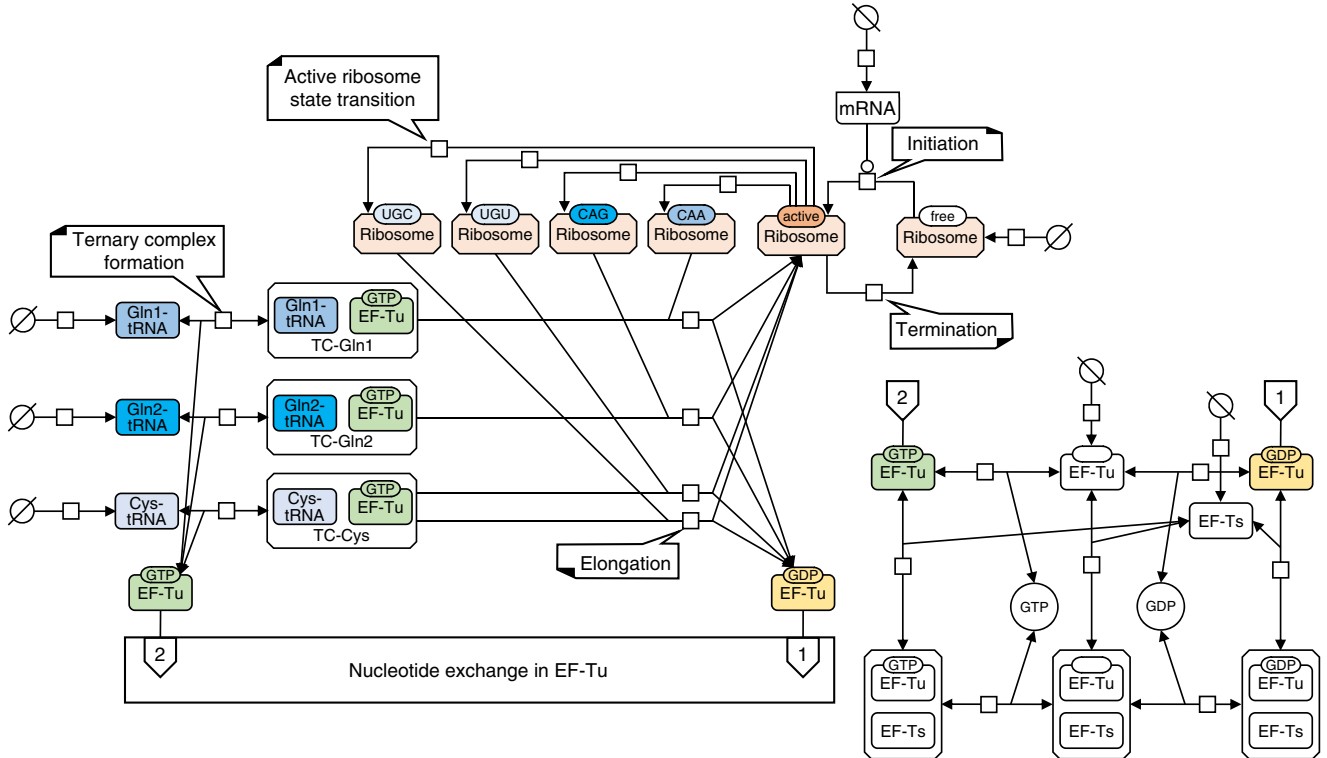

**Fig. 1 Schematic overview of the translation model.** A reduced pathway for elongation with amino acids cysteine (aminoacyl-tRNA Cys) and glutamine (aa-tRNA Gln1, Gln2) is represented in Systems Biology Graphical Notation. Initiation: free (unbound) ribosome gets converted to active ribosome, modulated by mRNA. Termination: active ribosome converts back to free ribosome at a rate fixed by the desired protein production rate. Active ribosome state transition: active ribosome instantaneously binds to codons (61 codons in full model, 4 here) at the fractions set by the specified proteome composition. Ternary complex formation: charged tRNAs (40 aa-tRNAs in full model, 3 here), replenished from a pool, combine with EF-Tu*GTP to form ternary complexes (40 TCs in the full model). Kinetic parameters of these reversible processes depend on the aa-tRNA. Elongation: labeled ribosome binds with the cognate TC to elongate the protein with the respective amino acid. The ribosome returns to its active state and EF-Tu*GDP is released. Other products of this reaction, such as deacylated tRNA, are not modeled. Nucleotide exchange (see right panel): EF-Tu*GDP is reactivated to EF-Tu*GTP in a sequence of steps modeled by reversible mass action kinetics. GTP and GDP pools are modeled with fixed concentrations. The nucleotide exchange is supported by EF-Ts, and the main flux is carried through the complexes formed by EF-Tu with EF-Ts.

across environments and growth rates in *E. coli*[14], as is the total mass concentration in the cytosol[15]. If the cell allocates more of this limited mass concentration budget to one particular process, less is available to other processes. The upper bound for the cytosolic mass concentration, beyond which diffusion becomes inefficient, is a fundamental constraint on cellular growth[16–18], and we thus use the cytosolic mass concentration of a particular molecule type as an approximation to its cost. Theoretical models of cellular growth that account for all major biochemical and biophysical constraints indicate that the limit on cellular dry mass indeed represents a dominant constraint on bacterial growth rates[12].

We hypothesize that to maximize the *E. coli* growth rate in a given environment, natural selection minimizes the total mass concentration of translation components utilized to achieve the required protein production rate. A corresponding optimality principle has been used to understand the relationship between the concentrations of enzymes and their substrates[19].

We find that a theoretical minimization of the combined cellular costs of the translation machinery components indeed leads to accurate predictions for their abundances, the resulting elongation rate, and the RNA/protein ratio. In addition to molecular masses, we also examine four alternative cost measures for cellular components that have been explored in the literature: (i) their protein content[9,20]; (ii) their carbon content[21]; and (iii) the energy[22,23] or (iv) the amount of catalysts[5,24] required for their production. We find that these alternative cost measures are strongly correlated for the studied components of the translation machinery and lead to very similar predictions for their abundances; the only cost measure that leads to substantially different predictions is the protein content, which does not assign any cost to tRNA and mRNA molecules.

## Results

**Cost minimization in a mechanistic model of translation.** To test our hypothesis, we constructed a translation model consisting of 274 biochemical reactions, including 119 reactions with non-linear kinetics. Figure 1 shows the modeled reactions for a subset of the 61 codons and the 40 species of charged tRNAs; for details see Methods, Supplementary Table 1, and Supplementary Data 1. This mechanistic model accounts for the concentrations of mRNA, the ribosome, the different charged tRNAs, and the elongation factors Ts (EF-Ts) and Tu (EF-Tu). We fully parameterized the model with molecular masses and kinetic constants measured experimentally[25–27]; the only exceptions are the translation initiation parameters, which were previously estimated from gene expression data[25], and the ribosomal Michaelis constant for the ternary complexes, which was previously estimated based on the diffusion limit[9]. The model is based purely on biochemical and biophysical considerations; it contains no adjustable parameters, nor does it include any explicit growth-rate dependencies.

For *E. coli* growing under different experimental conditions, we used measured growth rates and protein concentrations[28] to determine the required translation rate and the proportions of the different amino acids incorporated into the elongating proteins. At this required protein production rate, we minimized the combined cost of the translation machinery in our model, treating the concentrations of all components as free variables; the values of individual reaction fluxes result deterministically from these concentrations according to the respective rate laws (Methods). As the modeled kinetic rate laws are non-linear, all optimizations were performed numerically. In repeated optimization runs with two different solvers, we never found alternative optima, indicating that the optimization problem may be convex.

The results shown in the main text and figures are based on the assumption that costs are proportional to molecular masses; results based on other cost functions are shown in Supplementary Figs. 5 and 7.

**Predicted concentrations agree with observations.** We first compared our predictions to experimental data for exponentially growing *E. coli* in different conditions[8,28–31]. Figure 2 shows the results for growth in a glucose-limited chemostat at growth rate $\mu = 0.35\,\text{h}^{-1}$; for other conditions, see Supplementary Fig. 1. The mechanistic model accurately predicts the absolute concentrations of ribosomes, EF-Tu, EF-Ts, mRNA, and total tRNA in each condition. Predictions for individual tRNA concentrations are less accurate but are still mostly within a 2-fold error (Fig. 2, Supplementary Fig. 1); the discrepancies may be due to the simplifying assumption of the same ribosomal Michaelis constant $K_m$ for all tRNA species[9].

We next tested if this systems-level view on the total cost of translation explains the observed growth rate-dependencies of the expression of translation machinery components[8,28,29,31,32], of the elongation rate[32], and of the RNA/protein ratio[5,32], considering experimental data across 20 diverse conditions (14 minimal media, including 3 stress conditions; 4 chemostats; and 2 rich media)[28]. The predicted concentrations of ribosomes, EF-Tu, and EF-Ts increase with growth rate in line with experimental observations (Fig. 3).

Predicted absolute abundances of EF-Tu (Fig. 3a), EF-Ts (Fig. 3b), and mRNA (Supplementary Fig. 2a) account quantitatively for the experimental data[8,28–31], with average deviations (geometric mean fold-error) GMFE ≤ 27% for the elongation factors and GMFE = 6% for mRNA. At low growth rates, experimentally observed concentrations of EF-Tu (Fig. 3a) and tRNA (Supplementary Fig. 2b) are higher than predicted. The model only includes charged (aminoacyl-) tRNA concentrations, and it is likely that the unknown fraction of uncharged tRNA explains at least part of this deviation. Overall, the largest deviations between observed concentrations and predictions are seen in the two non-minimal conditions, which also exhibit the fastest growth ($\mu > 1\,\text{h}^{-1}$). A recent analytical study of balanced cellular growth indicates that these deviations may result from the influence of an increased growth-related dilution of cofactors and other intermediate metabolites, a phenomenon not included in our simulations[12].

**Active and de-activated ribosome fractions.** At low growth rates ($\mu < 0.3\,\text{h}^{-1}$; Fig. 3c), observed ribosome concentrations exceed those predicted from cost minimization, a deviation consistent with a substantial reserve of deactivated ribosomes at low growth rates[32]. Such deactivated ribosomes may provide fitness benefits in changing environments[33,34], but cannot be maximally efficient in a constant environment and thus cannot be predicted by our optimization strategy. To allow a meaningful comparison between predictions and experiment, we thus estimated the experimental concentration of ribosomes actively involved in elongation (Methods). Cost minimization predicts these experimental estimates with high accuracy across the full range of assayed growth rates; observed values deviate from predictions on average by GMFE = 14% (Fig. 3d).

The remaining, non-active ribosome fraction comprises two parts: the deactivated ribosome reserve currently unavailable for translation[32], and free, potentially active ribosomes not currently bound to mRNA (see Supplementary Note 2 for the nomenclature on ribosome states). As our model quantifies the abundance of both active and free ribosomes, their subtraction from observed total ribosome concentrations provides an

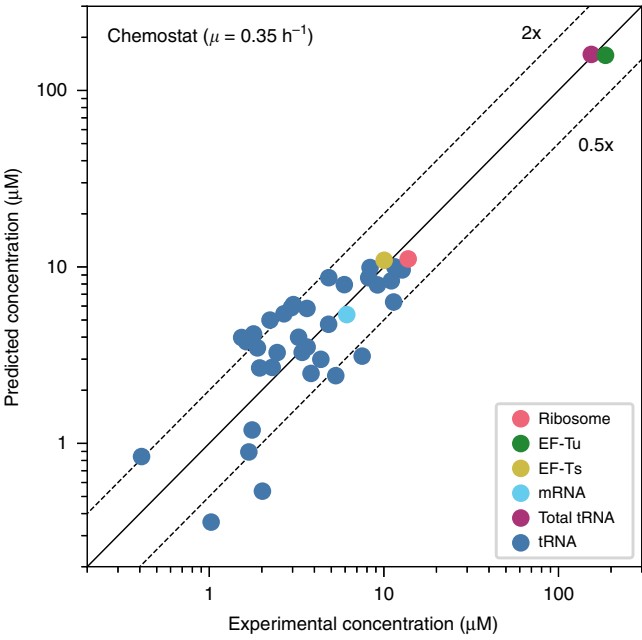

**Fig. 2 Predicted vs. observed concentrations in a glucose-limited chemostat.** Growth rate $\mu = 0.35\,h^{-1}$ (for other conditions, see Supplementary Fig. 1). The solid line shows the expected identity, whereas the upper and lower dashed lines show prediction errors of 2x and 0.5x, respectively. Predictions for ribosome, EF-Tu, EF-Ts, mRNA, and total tRNA are highly accurate, with Pearson's $R^2 = 0.99$ and geometric mean fold-error GMFE = 1.13, i.e., predictions based purely on a physico-chemical model and the assumption of cost minimization are on average 13% off. Predictions for individual tRNA species are somewhat less accurate, GMFE = 1.64. Experimentally determined concentrations of the ribosome (averaged over all ribosomal proteins), EF-Tu, and EF-Ts are from ref. [28]. mRNA[30] and tRNA[29] concentrations are interpolated values based on growth rates. Source data are provided as a Source Data file.

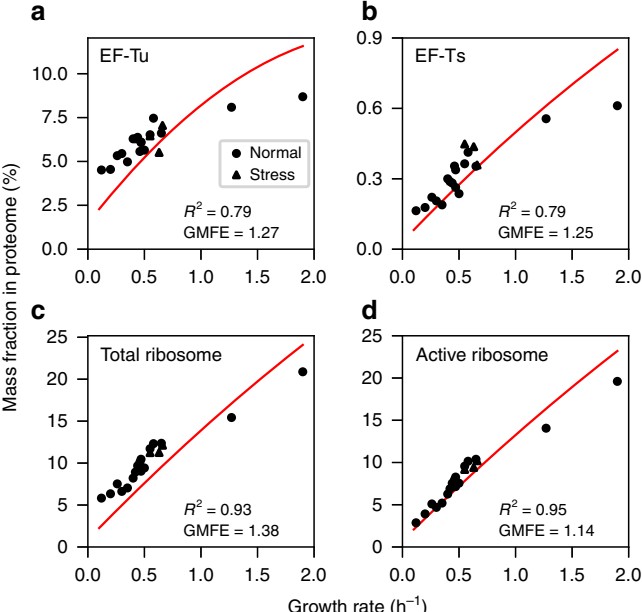

**Fig. 3 Growth rate dependence of predicted (red lines) and observed concentrations. a** EF-Tu, $R^2 = 0.79$, GMFE = 1.27. **b** EF-Ts, $R^2 = 0.79$, GMFE = 1.25. **c** Total ribosome concentration (arithmetic means across ribosomal proteins). **d** Actively elongating ribosomes, estimated from data in **c** according to ref. [32] (see Methods). Circles indicate normal conditions, triangles indicate stress conditions. Source data are provided as a Source Data file.

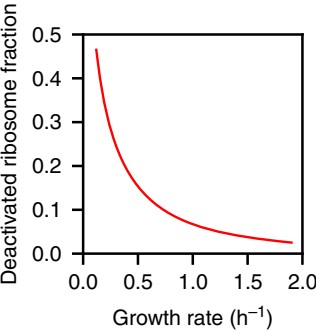

**Fig. 4 Estimated fraction of deactivated ribosomes.** The deactivated fraction reaches almost 50% for the lowest growth rate assayed in ref. [28] and drops rapidly towards zero at higher growth rates. Source data are provided as a Source Data file.

estimate of the deactivated ribosome reserve as a function of the growth rate (Fig. 4). While this reserve accounts for less than 20% of total ribosomes at fast to moderate growth, it reaches almost 50% at the lowest growth rate assayed in ref. [28].

**RNA/protein ratio and elongation rate.** A linear correlation between the RNA/protein ratio and growth rate was discovered in the 1950s[3,6,7,35] and forms the basis of phenomenological bacterial growth laws[5,9,32]. Relating the predicted total RNA (ribosomal RNA + tRNA + mRNA) with measured protein concentrations[28] indeed results in a near-linear relationship, accurately matching observed values at high to intermediate growth rates ($\mu > 0.3\,h^{-1}$; Fig. 5a). At lower growth rates, model predictions are slightly too low, likely because of the deactivated ribosome reserve[32] (Fig. 4). At low growth rates ($\mu = 0.12\,h^{-1}$), predictions of RNA and proteins allocated to an optimally efficient translation machinery (including deactivated ribosomes) account for 13% of total dry mass, rising almost linearly to 49% at high growth rates ($\mu = 1.9\,h^{-1}$; Supplementary Fig. 3).

The concentrations of the individual components of the translation machinery determine the average translation elongation rate (ribosomal activity), defined as the total cellular translation rate divided by the total active ribosome content[34]. The predicted elongation rates closely match the experimental data[32] over a broad range of growth rates (Fig. 5b).

**Cost minimization predicts response to antibiotics.** The expression of *E. coli*'s translation machinery reacts strongly to the

exposure to antibiotics that inhibit the ribosome, such as chloramphenicol[5,32,36]. The details of these changes can also be understood from our hypothesis of cost minimization. The concentrations of ribosomes and EF-Tu, the RNA/protein ratio, and the elongation rate of active ribosomes increase under chloramphenicol stress (Supplementary Fig. 4); these changes partially compensate for the reduced fraction of active ribosomes. The concentration of EF-Ts instead decreases with increasing chloramphenicol concentration (Supplementary Fig. 4c). EF-Ts contributes to translation by converting EF-Tu·GDP to EF-Tu·GTP, which then forms a ternary complex with charged tRNA. Under chloramphenicol stress, fewer ternary complexes are turned over, and hence less EF-Ts is needed.

**Alternative cost measures lead to similar results.** The biological fitness of *E. coli* cells depends on many factors, and hence any

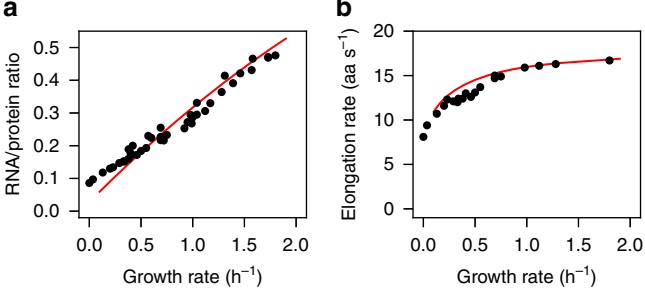

**Fig. 5 Growth rate dependences of total RNA/protein ratio and ribosome activity. a** Predicted total RNA concentration (mRNA + tRNA + rRNA) relative to observed total protein concentration at different cellular growth rates (red line) compared to experimental observations[5,32]; $R^2 = 0.97$, GMFE = 1.12. **b** Predicted (red line) and experimentally determined[32] elongation rates of actively translating ribosomes (ribosome activities); $R^2 = 0.93$, GMFE = 1.06. At the lowest assayed growth rates, non-growth-related translation—which is not included in the model—may become comparable to growth-related translation; at these growth rates, the numerical optimization of our model did not converge ($\mu < 0.1\,h^{-1}$), and thus the red lines are not extended into this region. Source data are provided as a Source Data file.

simple assignment of fitness costs to molecules can only be approximate. The results presented so far are based on the assumption that costs are proportional to molecular masses. To test if alternative cost measures lead to consistent results, we repeated our calculations using four distinct costs that have been employed in the literature.

Across most conditions, we obtained very similar predictions for the concentrations of translation machinery components when our model assigned molecular costs based on the carbon content of the molecules[21], on the amount of energy[22,23] spent on their production (ATP cost), or on the total investment into macromolecular catalysts[24] required for their production (synthesis cost) (Supplementary Figs. 5 and 6; we estimated ATP costs based on refs. [37,38], and calculated synthesis costs using flux balance analysis with molecular crowding[39–41], see Methods). All components whose concentrations we predict consist of protein, RNA, or both, and all costs examined are approximately proportional to the lengths of RNA and protein molecules. Thus, the relative costs of all components are essentially a function of the RNA/protein cost ratio $r$, i.e., the cost of RNA per nucleotide divided by the cost of protein per amino acid. We assume that the cost of RNA per nucleotide is identical for tRNA and rRNA; the corresponding cost ratio is broadly similar between molecular masses ($r = 3.0$), carbon content ($r = 2.0$), ATP cost ($r = 1.6$), and synthesis costs ($r = 1.7–2.1$) across minimal growth conditions ($\mu < 1\,h^{-1}$; Supplementary Fig. 7). In contrast, assuming that costs are proportional to only the protein content of the molecular assemblies[9,20] results in an RNA/protein cost ratio of zero. Predictions based on protein costs hence overestimate mRNA and tRNA concentrations (which cost nothing), resulting in corresponding underestimates of EF-Ts and especially EF-Tu concentrations (Supplementary Figs. 5, 6).

We note that in rich medium ($\mu = 1.9\,h^{-1}$), the RNA/protein cost ratio for synthesis costs is much lower than across minimal media, falling to $r = 0.21$ for tRNA and rRNA (Supplementary Fig. 7). This results in an overprediction of the observed tRNA concentration[8,29,31] by a factor of almost 2 (Supplementary Fig. 6). Moreover, while the predicted tRNA concentration is also almost twice the predicted EF-Tu concentration, experimental estimates for tRNA and EF-Tu are very similar[8,28,29,31,42]. We conclude that if the translation machinery has been optimized for

efficiency at high growth rates by natural selection, the synthesis cost of its components is unlikely to have been central to this optimization.

In sum, cost minimization in a mechanistic bottom-up model of optimal translation efficiency, fully parameterized with known kinetic constants and molecular masses, accounts quantitatively for the concentrations of all molecule species involved without any adjustable parameters. The optimal concentrations of different components change differentially with growth rate, explaining the observed scaling of *E. coli*'s translation machinery composition, RNA composition, and elongation rate. At least for the translation machinery part of the cellular economy, whose components consist largely of protein and RNA, approximate cost measures appear to be sufficient: several alternative cost measures provided predictions very similar to those based on molecular masses, emphasizing the generality of our findings.

We conclude that *E. coli*'s translation machinery works close to optimal cost efficiency. Accordingly, our findings are consistent with the hypothesis that natural selection has minimized a cost function similar to those examined here. Our results further support the idea that phenomenological growth laws of proteome composition[5,9,32,36] may have their root in the costs associated with the non-protein molecules involved in particular processes, and that their explicit inclusion in systems biology models of cellular growth[9,25,43,44] may eventually allow these models to abandon any reliance on phenomenological parameters.

## Methods

**Experimental concentrations of ribosomes, EF-Tu, and EF-Ts.** We used molar concentrations (μM) in the model; thus, all experimental data were converted to molar concentrations. We first calculated the total protein density during exponential growth on a glucose minimal medium at growth rate $\mu = 0.58\,h^{-1}$. In this condition, the total protein mass per cell is 280 fg (Supplementary Note 3 in ref. [28]), and cell volume is 1.90 fL (the cell volume 2.84 fL modified by a factor of 0.67 according to Supplementary Note 3 in ref. [28]). Accordingly, the protein mass density on glucose is $\rho_{P,glc} = 147.15\,g\,L^{-1}$.

We then fitted a second-order polynomial function $\phi(\mu)$ to the fraction of total protein in dry mass provided in ref. [1] across different growth rates $\mu$. With $\phi(\mu)$, $\rho_{P,glc} = 147.15\,g\,L^{-1}$ at $\mu = 0.58\,h^{-1}$, and the observed constant dry mass density of *E. coli* across growth conditions[14,15], we obtained the condition-specific total protein concentration, $\rho_P$, for all other growth conditions based on the respective observed growth rates.

For a given growth rate, $\mu'$, the total protein concentration $\rho_{P,\mu = \mu'}$ is given by

$$\rho_{P,\mu = \mu'} = \frac{\phi(\mu = \mu') \cdot \rho_{P,glc}}{\phi(\mu = 0.58)} \quad (1)$$

With the measured fraction of each protein ($f_{p,i}$) in the proteome[28], the molar concentration of each protein was then calculated as

$$c_i = \frac{\rho_P \cdot f_{p,i}}{MW_{p,i}}, \quad (2)$$

where $MW_{p,i}$ is the molecular weight of *protein i* in g·mole$^{-1}$. The ribosome concentration was calculated as the arithmetic mean of the molar concentrations of all ribosomal proteins.

**Experimental concentration of active ribosome.** Active ribosomes are defined here as ribosomes engaged in peptide elongation. Dai et al. estimated the fraction of active ribosomes, $f_{active}$, in *E. coli* at different growth rates[32]. We fitted a Michaelis-Menten type equation to their data, resulting in $f_{active} = \mu/(0.124 + \mu)$. For each total ribosome concentration $c_{ribosome}$ in ref. [28], we then estimated the corresponding active ribosome concentration as $c_{active-ribosome} = f_{active} \cdot c_{ribosome}$.

**Experimental concentrations of GTP and GDP.** GTP and GDP concentrations are from ref. [45]. We chose the data for growth on glucose for all simulations (See Supplementary Note 3 for details).

**Experimental concentration of mRNA.** mRNA concentration was calculated from the data given in ref. [30] as the ratio of mRNA copy number per cell and cell volume. To estimate the mRNA concentrations at the growth rates shown in Fig. 2 and Supplementary Fig. 1, we fitted a second order polynomial to the mRNA concentration as a function of growth rate; we then read off the values at the required growth rates. mRNA concentrations were assayed only at growth rates between

$0.11\,h^{-1}$ and $0.49\,h^{-1}$, and we did not attempt to extrapolate values beyond this range.

**Experimental concentration of tRNA**. We collected three independent datasets of tRNA concentrations. Dataset 1[29] contains tRNA concentration for each individual tRNA, whereas both dataset 2[8] and dataset 3[31] contain only total tRNA concentrations. In each of these experiments, tRNA abundance was measured as the ratio of tRNA to ribosomal RNA (rRNA). We scaled these values to absolute tRNA concentrations assuming that the rRNA concentration corresponds to the ribosome concentration estimated from the proteomics data (see the subsection "Experimental concentrations of ribosomes, EF-Tu, and EF-Ts"). To estimate the tRNA concentrations at the growth rates shown in Fig. 2 and Supplementary Fig. 1, we used the same fitting procedure as for mRNA.

**Concentrations of individual tRNAs and relationship to model**. Our model differentiates tRNAs by their anticodons (see below for details). Thus, 40 tRNAs were used to represent all elongator tRNAs. The tRNAs modeled in this work are listed in Supplementary Table 1 together with their common names used in dataset 1[29] and their gene IDs.

In the experiments by Dong et al. (dataset 1)[29], tRNAs were classified into 41 distinct sets based on two-dimensional polyacrylamide gel electrophoresis. We combined two tRNA sets corresponding to different tRNA weights if they have the same anticodon (i.e., the pairs of Val2A + Val2B, Thr1 + Thr3, and Tyr1 + Tyr2). The experimenters could not distinguish between the tRNAs Gly1 and Gly2, as these have very similar molecular weights and isoelectric point; the same was true for Ile1 and Ile2. We estimated the individual concentrations of these four tRNAs based on the ratios 3:2 between Gly1 and Gly2 and 20:1 between Ile1 and Ile2 observed by Ikemura et al.[46].

To estimate the tRNA concentrations at the growth rates shown in Fig. 2 and Supplementary Fig. 1, we fitted the concentration of each tRNA in dataset1 (measured for growth rates ranging from $0.28\,h^{-1}$ to $1.73\,h^{-1}$) to a second order polynomial of growth rate and extended this function to the required range, $0.12\,h^{-1}$ to $1.9\,h^{-1}$.

**Combination of tRNAs that are predicted to be non-expressed**. The predicted concentrations of 6 tRNAs are 0 μM; these are highlighted in red in Supplementary Table 2. This result is a straightforward consequence of the model structure. The relationship between codons and tRNAs is not one-to-one in *E. coli*. Consider a given codon (codon1) that has more than one cognate tRNA, say, tRNA1 and tRNA2. If tRNA2 is also the cognate tRNA of another codon (codon2), the predicted concentration of tRNA1 will be zero: for the same "price" (the same contribution to the limited total mass concentration), tRNA2 can service two codons, while tRNA1 can service only one. For example, codon GGG has two cognate tRNAs, gly1 and gly2; gly2 is also the cognate tRNA of codon GGA. Thus, both gly1 and gly2 can translate GGG, but gly2 can translate GGA, too, and is thus more valuable to the cell if we assume that both tRNAs are processed equally efficiently, as done in the model. Therefore, the predicted optimal concentration of gly1 will be zero (note that this might not occur in models that consider different ribosomal $k_{cat}$ or $K_m$ values for the two tRNAs). To compare our predictions to the experimental data[29], we combined tRNAs with predicted zero concentration with their co-functioning tRNAs in both the predictions and the experimental data. The resulting six combined tRNA pairs are: GLy1 + Gly2; Leu1 + Leu3; Leu4 + Leu5; Pro1 + Pro3; Ser2 + Ser1; Thr2 + Thr4. In all reported figures, the total number of tRNAs shown is thus 34.

**Concentrations of amino acids and total protein**. We first calculated a typical molecular protein mass for each condition. $MW_p$ is the weighted average over the molecular masses of all proteins assayed by Schmidt et al., where the weights are the corresponding proteome fractions in that condition.The molar concentration of protein, $c_{protein}$, is given by

$$c_{protein} = \frac{\rho_p}{MW_p} \tag{3}$$

where $\rho_p$ is the mass concentration of total protein at the given condition (Eq. 1). The concentration of amino acids encoded by *codon-i* is given by:

$$c_{codon-i} = f_{codon-i} \cdot L_{protein} \cdot c_{protein} \tag{4}$$

where $L_{protein}$ is the abundance-weighted average protein length at the given condition; $f_{codon-i}$ is the frequency of codon $i$ in the genome, where each gene is weighted by its relative abundance in the proteome.

Note that for an amino acid $AA_j$ encoded by multiple synonymous codons, $c_{codon-i}$ is not the total concentration of $AA_j$ in cellular proteins, but only of the fraction encoded by *codon-i*; the total concentration of $AA_j$ is obtained by summing the $c_{codon-i}$ values for all synonymous codons for $AA_j$.

For simulations under chloramphenicol stress, $c_{protein}$ and $c_{codon-i}$ are not available. We approximated their values by the corresponding concentrations for growth on glucose in the absence of the antibiotic.

**Mass fraction of translation machinery in total dry weight**. The dry mass fraction of the translation machinery shown in Supplementary Fig. 3 includes ribosome, mRNA, charged tRNAs, EF-Tu, and EF-Ts; it does not include GDP, GTP, free tRNA, tRNA-synthetases, and elongation factor G (FusA). We converted from protein fractions to mass fractions of total dry weight using the relationship between total protein mass and dry mass discussed in the subsection "Experimental concentrations of ribosomes, EF-Tu, and EF-Ts".

*Experimental estimate:* The mass fraction of the translation machinery in total dry weight is the sum of two parts: (1) protein and (2) RNA.

(1) We calculated the mass fraction of translational proteins (including ribosomal protein, EF-Tu, and EF-Ts) in dry weight from the proteomics data in ref. [28].

(2) We fitted the reported total RNA/protein ratio in refs. [5,32] to a second order polynomial of growth rate. We then used this fitted function to calculate the RNA/protein ratio at the growth rates assayed by Schmidt et al.[28], and multiplied this ratio with the dry mass fraction of protein.

*Theoretical prediction*: The predicted dry mass fraction of the translation machinery was the ratio of the total mass concentration of the translation machinery (including free ribosome, active ribosome, EF-Tu, EF-Ts, charged tRNA, and mRNA) to the total dry mass density. The total dry mass density was estimated as $\rho_{p,glc}/\phi(0.58\,h^{-1}) = 147.15\,g\,L^{-1}/0.631 = 233.30\,g\,L^{-1}$. For the prediction including de-activated ribosome concentrations (dashed line in Supplementary Fig. 3), we added estimates of de-activated ribosome concentrations according to Fig. 4 of the main text.

**Molecular weights**. Molecular weights of ribosome, charged tRNAs (aa-tRNAs), EF-Tu, EF-Ts, and the ternary complexes (TC, EF-Tu·GTP·aa-tRNA) were calculated from their sequences. The stoichiometry of ribosomal proteins and RNAs in the ribosome was obtained from the EcoCyc database[47]; the stoichiometry of all components is 1 except for RplL, for which it is 4.

We used an average mRNA to represent the total mRNA. The molecular weight of an average mRNA ($MW_{mRNA}$) is the sum of two parts: (1) the molecular weight of the coding sequence (CDS) of mRNA ($MW_{mRNA-CDS}$), which was calculated from protein-expression-weighted mRNA length and nucleotide composition of *E. coli* protein-coding sequences in each growth condition[28]; (2) the weight of the untranslated region (UTR) of mRNA ($MW_{mRNA-UTR}$), which was calculated from the average nucleotide composition of the genome and the typical length of UTR. The length of UTR was assumed to be 85 nt, a typical length of the untranslated region in *E. coli*[48]. Thus,

$$MW_{mRNA} = MW_{mRNA-CDS} + MW_{mRNA-UTR}. \tag{5}$$

In the simulations of translation under antibiotic stress, the molecular weight of chloramphenicol was set to 0.

**Alternative costs of translation machinery components**. The hypothesis underlying our analysis is that the components of the translation machinery are expressed to minimize the total cost of translation at a given protein production rate. For the calculations underlying the figures of the main text, we assumed that molecular costs are proportional to molecular mass concentrations. To test alternative cost measures proposed in the literature, we estimated the costs of each translation machinery component in terms of (1) its carbon content[21] (carbon cost); (2) the total number of high-energy phosphate bonds required for its production (ATP cost); (3) the total enzyme mass required for its production (synthesis cost)[24]; and (4) its protein content[9,20] (protein content). Some of these cost measures are condition-dependent. The estimated costs are provided in Supplementary Data 2; the RNA/protein cost ratios are compared in Supplementary Fig. 7 and listed in Supplementary Table 3. All costs were estimated per component (i.e., per molecule or per macromolecular complex). Note that all components whose costs are considered in the model consist of protein, RNA, or both.

**Carbon cost as an alternative cost measure**. The carbon cost of a component is its total number of carbon atoms.

**ATP cost as an alternative cost measure**. The ATP cost of a component is the number of high-energy phosphate bonds (denoted ~P) that were invested into its production. The ATP costs includes (1) the ATP invested into the synthesis of the precursors (nucleoside triphosphates or amino acids) and (2) the ATP cost of polymerization during RNA transcription or protein translation.

*The ATP cost of amino acid synthesis*: For cells growing on minimal carbon media, the ATP cost of amino acid production was obtained from ref. [37]. Since the ATP costs of a given amino acid are very similar across minimal media with different carbon sources[37], we used the ATP costs for amino acid synthesis on glucose for all minimal media considered. For *E. coli* growing on glycerol + amino acids[28], the ATP production cost for amino acids was assumed to be zero.

*The polymerization cost of protein*: The polymerization cost of protein is 4 ATP per amino acid: two ATP for tRNA charging, 1 ATP for EF-Tu in elongation, and 1 ATP for elongation factor G (EF-G) in elongation.

*The ATP cost of NTP synthesis*: For cells growing on minimal carbon media, the de novo synthesis cost of NTP was obtained from ref. [38]. The glycerol + amino acids medium used in the proteomics study[28] also contains adenine and uracil; here, we assumed that the synthesis of ATP and GTP starts from adenine and that the synthesis of UTP and CTP starts from uracil. PRPP (5-phospho-α-D-ribose 1-diphosphate), whose production consumes 29 ATP[37], was considered to be the donor of ribose to the synthesis of NTPs. Finally, we estimated the total energy (~P) costs in the glycerol + amino acids medium for ATP, UTP, CTP, and GTP as 31, 31, 29, and 32, respectively. We did not attempt to estimate the ATP cost of NTP synthesis in the LB condition, as it is not clear to what extent NTPs are taken up from the medium.

*The polymerization cost of RNA*: The polymerization cost of RNA is 0 ATP per base, as no high-energy phosphate bonds beyond those of the polymerized NTPs are required.

*The degradation cost of mRNA*: The degradation rates of tRNA and rRNA are much lower than their production rates, and hence we did not account for their degradation. In contrast, mRNA is degraded much more quickly than tRNA and rRNA, and we thus considered the influence of degradation on the mRNA polymerization cost. At steady state, all degraded mRNA (in the form of nucleoside mono-phosphates, NMPs) is assumed to be recycled to re-transcribe mRNA. These recycled NMPs require two ~P to form NTPs, so the mRNA recycling cost ($cost_{NTP-deg}$) is 2 ~P per NTP. At a given mRNA concentration $c_{mRNA-NTP}$ (in units of NTPs built into mRNA), the production of mRNA must offset the combination of mRNA degradation and mRNA dilution by cellular growth at rate $\mu$. The rate of ATP consumption for this production is thus given by

$$v_{cost-ATP-mRNA} = \mu \cdot cost_{NTP} \cdot c_{mRNA-NTP} + k_{deg} \cdot cost_{NTP-deg} \cdot c_{mRNA-NTP} \quad (6)$$

where $cost_{NTP}$ is the synthesis cost of NTPs (estimated above, "The ATP cost of NTP synthesis") and $k_{deg}$ is the mRNA degradation rate constant. $k_{deg}$ is calculated from mRNA half-life ($t_{half}$), $k_{deg} = \ln(2)/t_{half}$, with $t_{half} = 5$ min for all growth conditions[49]. To obtain the ATP cost per NTP in mRNA, we must divide this rate by $\mu \cdot c_{mRNA-NTP}$:

$$cost_{NTP-mRNA} = \frac{v_{cost-ATP-mRNA}}{\mu \cdot c_{mRNA-NTP}} = cost_{NTP} + \frac{k_{deg}}{\mu} \cdot cost_{NTP-deg} = cost_{NTP} + \frac{2k_{deg}}{\mu} \quad (7)$$

Thus, mRNA degradation dominates the ATP cost of mRNA at very low growth rates, but becomes insignificant at growth rates higher than the mRNA degradation rate.

**Synthesis cost as an alternative cost measure**. The synthesis cost is the total macromolecular dry mass, which includes transporters, enzymes, RNA polymerase, and ribosome, that is needed to synthesize each component of the translation machinery. To estimate the synthesis cost of each component, we first estimated the macromolecular dry mass that is needed to synthesize one millimole of amino acid ($cost_{AA}$) and one millimole of nucleotide ($cost_{nucl}$). Based on these estimates, the synthesis cost of a protein is $cost_{protein} = L_{protein} \cdot cost_{AA}$, where $L_{protein}$ is the protein length in amino acids, and the synthesis cost of an RNA molecule is $cost_{RNA} = L_{RNA} \cdot cost_{nucl}$, where $L_{RNA}$ is the length of the RNA molecule in nucleotides.

We calculated $cost_{AA}$ and $cost_{nucl}$ using ccFBA[40,41], which is an implementation of the MOMENT[39] algorithm for flux balance analysis with molecular crowding, featuring an improved treatment of co-functional enzymes. Briefly, ccFBA assigns each enzyme a constant catalytic rate ($k_{cat}$) and molecular weight, and then finds the flux distribution that maximizes biomass production while not exceeding a threshold on the total enzyme mass.

*Synthesis cost of protein*: We first added a protein synthesis reaction to the iML1515 model implemented in ccFBA. The stoichiometric coefficient of each amino acid consumed in this reaction was set to its proportion in the biomass reaction of the iML1515 model. In *E. coli*, there are approx. 9 TCs per ribosome at high growth rates[29,42]; we thus designated the ribosome plus 9 ternary complexes (TCs) as the "enzyme" of the protein synthesis reaction. We parameterized this "enzyme" with $k_{cat} = 22$ s$^{-1}$ and molecular weight = 2933.241 kD[1]. We then set the objective function to the rate of protein production, $v_{AA}$, instead of the biomass production rate $v_{bio}$. We simulated different growth conditions by only allowing the model to import nutrients available in the respective medium, setting the lower bound of the corresponding exchange reactions to $-1000$ (mmol·g$_{DW}^{-1}$·h$^{-1}$). For cells growing on LB medium, the lower bound of all exchange reactions was set to -1000, i.e., all metabolites for which there is a transporter in the model can be taken up. We maximized the protein synthesis rate $v_{AA}$, given a limit on the dry mass fraction of macromolecules involved in metabolism (a "budget") of C = 0.27. As the lower bound chosen for the exchange rates was very high, the optimizations were constrained by C, i.e., the optimal $v_{AA}$ value is the maximal rate of protein production with this macromolecular budget. Accordingly, the synthesis cost of amino acids is $cost_{AA} = 0.27/v_{AA}$, expressed as the dry mass fraction required to produce 1 mmol·g$_{DW}^{-1}$·h$^{-1}$ of amino acids. The computed protein synthesis costs for all 20 conditions considered are shown in Supplementary Table 4.

*Synthesis cost of stable RNA*: To estimate the cost per nucleotide of stable RNA production, $cost_{nucl}$, we implemented an analogous algorithm. We first added an RNA synthesis reaction to the iML1515 model[50] and set this reaction as the objective function. The stoichiometry of NTPs consumed in this reaction was set to the corresponding fractions of NTPs in the biomass reaction of the iML1515

model. We designated the RNA polymerase (molecular weight = 389.11 kD) as the enzyme catalyzing this reaction. For stable RNA (tRNA and rRNA) synthesis, the turnover rate of the RNA polymerase (RNA-P)[1] is $k_{RNAP-sRNA} = 85$ s$^{-1}$. We maximized the flux of the RNA synthesis reaction, $v_{sRNA}$, constrained by the macromolecular budget C = 0.27. The synthesis cost of stable RNA per nucleotide was then calculated as $cost_{nucl-sRNA} = 0.27/v_{sRNA}$, expressed as the dry mass fraction required to produce 1 mmol·g$_{DW}^{-1}$·h$^{-1}$ of NTP in stable RNA. The computed stable RNA synthesis costs for all 20 conditions considered are listed in Supplementary Table 4.

*Synthesis cost of mRNA*: In *E. coli*, mRNA transcription is slower than stable RNA transcription. The turnover rate of RNA-P for mRNA is $k_{RNAP-mRNA} = 66$ s$^{-1}$ (i.e., the fastest rate of mRNA transcription has been observed to be about three times the maximal translation rate)[51]. As for the calculation of the ATP cost of mRNA, we need to account for mRNA degradation at rate $k_{deg}$ (which is much faster than the degradation of stable RNA). We again assumed that nucleotides from degraded mRNA are re-used by RNA-P to synthesize mRNA. As before, we note that at a given mRNA concentration $c_{mRNA}$, the production of mRNA must offset the combination of mRNA degradation and mRNA dilution by cellular growth at rate $\mu$. The rate of mRNA production by the RNA-P is thus

$$v_{mRNA} = \mu \cdot c_{mRNA} + k_{deg} \cdot c_{mRNA} \quad (8)$$

To obtain the concentration of RNA-P necessary to catalyze this rate, we need to divide this expression by $k_{RNAP-mRNA}$:

$$c_{RNAP-mRNA} = \frac{v_{mRNA}}{\mu} = c_{mRNA}\left(1 + \frac{k_{deg}}{\mu}\right) \quad (9)$$

Thus, the concentration of RNA-P required for mRNA production is larger by a factor $(1 + k_{deg}/\mu)$ when accounting for mRNA degradation than it would be otherwise. To account for mRNA degradation, we thus set the effective turnover number of RNA-P to $k_{eff-mRNA} = k_{RNAP-mRNA}/\left(1 + k_{deg}/\mu\right)$. We maximized the flux of the mRNA synthesis reaction, $v_{mRNA}$, constrained by the macromolecular budget C = 0.27. The synthesis cost of mRNA per nucleotide was then calculated as $cost_{nucl-mRNA} = 0.27/v_{mRNA}$, expressed as the dry mass fraction required to produce 1 mmol g$_{DW}^{-1}$ h$^{-1}$ of NTP in mRNA. The computed costs for all 20 conditions considered are provided in Supplementary Table 4.

**Protein content as an alternative cost measure**. The protein content was considered as the only relevant cost of the translation machinery in two previous models[9,20]. These models accurately predicted the ribosomal protein fraction in total protein, whereas they substantially underestimated the EF-Tu proteome fraction compared to measurements data[9,20].

Following these earlier works, we calculated the protein cost of a component as the number of amino acid residues it incorporates. Since mRNA and tRNA have zero protein content, this definition assigns no costs to their expression, potentially leading to a prediction of infinite concentrations. To avoid such pathological predictions, we set the protein content costs of mRNA and charged tRNAs to small, arbitrary values in the model (mRNA: 10 amino acid residues; tRNA: 5 amino acid residues).

**Model overview**. The mechanistic translation model encompasses the processes of translation initiation, elongation, termination, nucleotide exchange in EF-Tu, and ternary complex (TC) formation. Figure 1 of the main text illustrates the modeled reaction; for better readability, the figure shows only a subset of codon/charged tRNA combinations. In total, the model includes 274 reactions.

**Translation initiation**. During initiation, mRNA converts free ribosomes to active ribosomes:

$$\text{Ribosome}_{free} \rightarrow \text{Ribosome}_{active}. \quad (r1)$$

Translation initiation consists of multiple elementary reactions[52]. However, a recent study found that at steady state, the kinetics of initiation effectively follow Michaelis–Menten kinetics, with mRNA in the enzyme position (with concentration $c_{mRNA}$) and free (unbound) ribosomes in the substrate position (with concentration $c_{ribo-free}$)[53],

$$v_{tl-init} = k_{cat-mRNA} \cdot c_{mRNA} \cdot \frac{c_{ribo-free}}{K_{M-ribo} + c_{ribo-free}} \quad (10)$$

with $k_{cat-mRNA} = 1.33$ s$^{-1}$ and $K_{M-ribo} = 8.5$ μM from ref. [25].

We assume that the turnover number of mRNA for ribosome binding ($k_{cat-mRNA}$) is growth rate-independent, and hence that the observed growth rate-dependent activity is due to changes in the concentration of free ribosomes available for initiation[54]. Thus, we used the maximal reported mRNA activity as an estimate of $k_{cat-mRNA}$.

**Ternary complex formation and EF-Tu nucleotide exchange**. Ternary complex formation and nucleotide exchange in EF-Tu are the processes by which the translation machinery recycles its substrates, the ternary complexes, for elongation.

*TC formation*: The binding of EF-Tu·GTP to charged-tRNA (aa-tRNA) forms the ternary complex (*ternary complex formation* in Fig. 1), which is the substrate of translation elongation.

*Nucleotide exchange in EF-Tu*: EF-Tu·GDP is released after the formation of a new peptide bond. Elongation factor Ts (EF-Ts) binds to EF-Tu·GDP and induces the exchange of GDP for GTP (right panel in Fig. 1).

The individual steps of these two processes are modeled with mass action kinetics[26,27].

In the implementation of the model, we divide each reversible reaction into two irreversible reactions. The TC formation reaction is a set of reactions that include the binding of 40 aa-tRNAs to Tu·GTP, and its rate constants depend on which amino acid is involved[27] (for the parameter values see Supplementary Data 1).

**Elongation**. Elongation is a very complex process[55,56]. For simplicity, we model elongation as a single reaction with an active ribosome as the enzyme and TC as the substrate, which was proposed by Klumpp et al.[9]. In this single reaction model, TC is discriminated by the anticodon and all TCs are treated with the same activity. Michaelis–Menten kinetics are used to describe the reaction rate[9]. There are 40 anticodons in total for all elongator tRNAs in *E. coli*; accordingly, our model uses 40 tRNAs to represent all tRNAs.

At steady state, the total translation rate (per cytosolic volume) of each codon remains constant. We do not model the translation of a whole protein. Instead, we decompose protein synthesis into the translation of 61 codons (see also "Modeling" below).

*Active ribosome state*: We distinguish active ribosomes according to their binding codons (codon in ribosome A site). Thus, there are 61 types of active ribosome in the model, distinguished in the model by subscripts indicating the codon currently presented by the ribosome:

$$\text{Ribosome}_{active} \rightarrow \text{Ribosome}_{codon-i} \quad (r2)$$

here, *codon-i* is one of the 61 codons and Ribosome$_{codon-i}$ is the active ribosome that binds *codon-i*. This reaction is constrained by mass balance, but is considered to be instantaneous. We assume that when an active ribosome binds with a specific codon, it only translates the codon's cognate tRNA.

In the model, there are 61 codons and 40 tRNAs, and the relation between tRNA and codon is not one-to-one. Based on the number of cognate tRNAs, we partition the 61 codons into 2 classes: class 1 codons have one cognate tRNA, whereas class 2 codons have two cognate tRNAs. The lists of class 1 and class 2 codons are provided in Supplementary Table 5.

For class 1 codons ($n = 51$), the elongation reaction is:

$$\text{TC}_{codon-i} \overset{\text{Ribosome}_{codon-i}}{to} \text{EF} - \text{Tu} \cdot \text{GDP} + \text{tRNA}_{codon-i} + \text{aa}_{codon-i}, \quad (r3)$$

where TC$_{codon-i}$ is the cognate TC of *codon-i*, tRNA$_{codon-i}$ is the released free tRNA, and aa$_{codon-i}$ symbolizes the amino acid that was just appended to the growing peptide. The tRNA$_{codon-i}$ and aa$_{codon-i}$ are included here for completeness, but are not included explicitly in the optimized model, as they do not influence the results once appropriate exchange reactions have been added. Simultaneously, the Ribosome$_{codon-i}$ is converted to Ribosome$_{active}$, which is ready to participate in the next round of elongation,

$$\text{Ribosome}_{codon-i} \rightarrow \text{Ribosome}_{active}. \quad (r4)$$

For simplicity, we combine r3 and r4 into the new reaction r5, with Ribosome$_{codon-i}$ as a substrate and Ribosome$_{active}$ as a product (the same for r6 and r7),

$$\text{TC}_{codon-i} + \text{Ribosome}_{codon-i} \overset{\text{Ribosome}_{codon-i}}{\rightarrow} \text{Ribosome}_{active} + \text{EF} - \text{Tu} \cdot \text{GDP}$$
$$+ \text{tRNA}_{codon-i} + \text{aa}_{codon-i}. \quad (r5)$$

The translation rate of *codon-i* is described by Michaelis–Menten kinetics,

$$v_{tl-codon-i} = c_{ribo-codon-i} \cdot k_{\text{cat-ribo}} \cdot \frac{c_{TC-codon-i}}{c_{TC-codon-i} + K_{M-TC}}, \quad (11)$$

where $c_{ribo-codon-i}$ is the concentration of ribosomes that present *codon-i* (Ribosome$_{codon-i}$), $c_{TC-codon-i}$ is the concentration of cognate TC of *codon-i* (TC$_{codon-i}$), $k_{\text{cat-ribo}} = 22\ \text{s}^{-1}$, and $K_{M-TC} = 3\ \mu\text{M}$ (parameters from ref. [9]).

For class 2 codons ($n = 10$), the active ribosome can translate two TCs and thus there are two reactions:

$$\text{TC}_{codon-i-1} + \text{Ribosome}_{codon-i} \overset{\text{Ribosome}_{codon-i}}{\rightarrow} \text{Ribosome}_{active} + \text{EF} - \text{Tu} \cdot \text{GDP}$$
$$+ \text{tRNA}_{codon-i-1} + \text{aa}_{codon-i}. \quad (r6)$$

and

$$\text{TC}_{codon-i-2} + \text{Ribosome}_{codon-i} \overset{\text{Ribosome}_{codon-i}}{\rightarrow} \text{Ribosome}_{active} + \text{EF} - \text{Tu} \cdot \text{GDP}$$
$$+ \text{tRNA}_{codon-i-2} + \text{aa}_{codon-i}. \quad (r7)$$

The translation rate of *codon-i* is the sum of these two reactions:

$$v_{tl-codon-i} = v_1 + v_2, \quad (12)$$

with

$$v_1 = c_{ribo-codon-i} \cdot k_{\text{cat-ribo}} \cdot \frac{c_{TC-codon-i-1}}{(c_{TC-codon-i-1} + c_{TC-codon-i-2}) + K_{M-TC}} \quad (13)$$

and

$$v_2 = c_{ribo-codon-i} \cdot k_{\text{cat-ribo}} \cdot \frac{c_{TC-codon-i-2}}{(c_{TC-codon-i-1} + c_{TC-codon-i-2}) + K_{M-TC}}. \quad (14)$$

**Termination**. In termination, an active ribosome is converted to a free ribosome:

$$\text{Ribosome}_{active} \rightarrow \text{Ribosome}_{free}. \quad (r8)$$

The termination rate is equal to the protein synthesis rate at steady state:

$$v_{\text{term}} = v_{\text{protein-syn}} = \mu \cdot c_{\text{protein}}, \quad (15)$$

where $c_{\text{protein}}$ (Eq. 3) is the absolute concentration of protein measured experimentally.

**Modeling exchange reactions**. Besides the reactions mentioned above, we also add exchange reactions that allow the influx of free ribosome, charged tRNA, mRNA, EF-Tu, EF-Ts, and GTP into the system. We also add exchange reactions that allow efflux of GDP out of the system.

**Modeling antibiotic stress**. To model chloramphenicol (cm) stress, we add the exchange reaction for chloramphenicol. All forms of ribosome can be inhibited by chloramphenicol:

$$\text{Ribosome} + \text{cm} \leftrightarrow \text{Ribosome} \cdot \text{cm}, \quad (r9)$$

where Ribosome includes both free and active ribosomes. The reaction rate is described by mass action kinetics with $k_{\text{on}} = 0.00057\ \mu\text{M}^{-1}\ \text{s}^{-1}$ (0.034 $\mu\text{M min}^{-1}$) and $k_{\text{off}} = 0.0014\ \text{s}^{-1}$ (0.084 min$^{-1}$)[57].

For simplicity, we make the following assumptions:

(1) chloramphenicol diffuses freely across cell membranes, such that the intracellular concentration of free chloramphenicol is the same as that in the medium;

(2) chloramphenicol bound to an active ribosome (ribosome·cm) causes the ribosome·cm complex to dissociate quickly from the mRNA, thus not affecting further translation of the mRNA.

We assume that the reaction of ribosome and chloramphenicol binding is at steady state (dynamic equilibrium), and thus the active ribosome concentration will be constant. However, not all active ribosomes will be able to successfully finish translation, as the active ribosome can be inhibited by chloramphenicol during translation. Thus, to estimate the production rate of functional proteins, we need to estimate the probability that the active ribosome can finish translation without chloramphenicol inhibition.

To calculate this probability, we use the method proposed in ref. [32]. The probability of chloramphenicol binding to a ribosome in a given time unit is:

$$k_{\text{hit}} = k_{\text{on}} c_{\text{cm}}, \quad (16)$$

where $k_{\text{on}}$ is the binding constant. The probability that the ribosome is bound $n$ times in the time interval $t$ follows the Poisson distribution

$$P(n) = e^{-k_{\text{hit}} t_{\text{tl}}} \frac{k_{\text{hit}} t_{\text{tl}}}{n!}, \quad (17)$$

where $t_{\text{tl}}$ is the experimental measured translation time of the translated gene. The probability that the ribosome can finish translation without inhibition by chloramphenicol is $P(0)$:

$$P_{\text{tl}} = e^{-k_{\text{hit}} t_{\text{tl}}}. \quad (18)$$

Here, $t_{\text{tl}}$ is the time for *LacZ* and $t_{\text{tl}} = 72$ s (from ref. [32]). For simplicity, we assume that all codon-presenting active ribosomes (61 forms of active ribosome) have the same $P_{\text{tl}}$, and thus the effective concentration of the ribosome presenting *codon-i* is

$$c_{ribo-eff-i} = P_{\text{tl}} c_{ribo-i}. \quad (19)$$

Under inhibition by chloramphenicol, this effective ribosome concentration of *codon-i* ($c_{ribo-eff-i}$) replaces the ribosome concentration of ($c_{ribo-i}$) in the model.

**Model optimization**. We assume translation at steady state and use a constraint-based optimization model. The constraints are given by the above equations and by the requirement of a given total cellular rate of protein synthesis (estimated as the product of growth rate and experimental proteome composition at this growth rate). For simplicity, we decomposed protein synthesis into the translation of 61 codons, and so the constraint on protein synthesis rate is implemented as 61 individual equations, each representing the translation rate of one codon. At steady state, the codon translation rate equals the dilution rate of amino acids incorporated at protein positions encoded by *codon-i*, i.e., the growth rate $\mu$ multiplied by the concentration of amino acids coded by *codon-i* in proteome data:

$$v_{tl-codon-i} = \mu \cdot c_{codon-i}, \quad (20)$$

note that if multiple codons encode the same amino acids, then the total concentration of that amino acid in cellular proteins is the sum over the $c_{codon-i}$ values for the individual codons.

Given these constraints, we minimize the total mass concentration of the modeled translation apparatus, consisting of ribosome, EF-Tu, EF-Ts, mRNA,

GTP, GDP, and charged tRNAs (aa·tRNA),

$$\sum_{m \in C} \mathrm{MW}_m \cdot c_m, \qquad (21)$$

where the molecule types together form the set $C$, $\mathrm{MW}_m$ is the molecular weight, and $c_m$ is the concentration of molecule type $m$. We do not minimize the contributions of GTP and GDP, who participate in multiple other cellular processes[58] and whose concentrations are thus unlikely to be dominated by translation; their concentrations are consequently fixed to experimentally observed values in our simulations (see Supplementary Note 3).

Let $R$ be the set of reactions that together comprise translation. We also consider the dilution of the molecules involved in these reactions due to cellular volume growth at rate $\mu$:

$$\mathrm{S} \cdot \mathrm{v}(\mathbf{c}) - \mu \cdot \mathbf{c} = 0, \qquad (22)$$

where $\mathbf{S}$ is the stoichiometric matrix for the reactions in $R$, $\mathbf{c}$ is a vector of the concentrations $c_m$, and $\mathbf{v}(\mathbf{c})$ is the corresponding vector of reaction rates $v_i$, with the concentration-dependent kinetics described above.

Thus, we solve the non-linear constrained optimization problem:

$$\min_{\mathbf{c}} \sum_{m \in C} \mathrm{MW}_m \cdot c_m. \qquad (23)$$

subject to:

$\mathrm{S} \cdot \mathrm{v}(\mathbf{c}) - \mu \cdot \mathbf{c} = 0$
$v_{tl-codon-i} = \mu \cdot c_{codon-i}$ for i=1, …, 61
$v_{\mathrm{term}} = \mu \cdot c_{\mathrm{protein}}$

We formulated the optimization problem in GAMS and used the BARON global solver[59] on NEOS Server[60] with 8 h as the time limit to solve this problem. Because the problem is non-linear, it is not clear a priori if it is convex, in which case only a single optimum would exist. The problem is conceptionally similar to the one studied by Noor et al.[24], and thus convexity is conceivable. A first optimum was returned by the solver within at most a few minutes for all optimizations performed. However, to guard against the existence of overlooked alternative optima, we allowed the search to continue for a total of 8 h in each case. In addition, we repeated all simulation with the global non-linear solver in Lingo 13 (LINDO Systems, Inc., https://www.lindo.com/index.php/products/lingo-and-optimization-modeling). No alternative optima were ever found; we thus have no evidence of non-convexity.

To assess the alternative costs, the molecular weight of each component was replaced by the corresponding cost measure (see the section "*Alternative costs of translation machinery components*"). The costs for each component are shown in Supplementary Data 2, and the corresponding predictions are shown in Supplementary Data 3.

**Reporting summary**. Further information on research design is available in the Nature Research Reporting Summary linked to this article.

## Data availability
The experimental data that support the findings of this study are available from the original publications[5,8,28–32,36]. Source data are provided with this paper.

## Code availability
Model: The optimization problem, including the model and its parameterization, is provided as an SBML file (Supplementary Data 4) and as a GAMS input file (Supplementary Data 5, with protein production requirements set to those for growth on minimal glucose medium). In addition, the model has been submitted to Biomodels (MODEL2006210001). GAMS (General Algebraic Modeling System, https://www.gams.com/) is a modeling platform for mathematical optimization, it provides dozens of solvers for a variety of optimization problems. The model is written in GAMS (version 25.1.2). BARON (Branch-And-Reduce Optimization Navigator) is a computational system for solving nonconvex optimization problems to global optimality. NEOS Server is a free online server that provides state-of-the-art solvers for numerical optimization. We used the BARON solver (version 19.12.7) on NEOS Server (Version 5.0) for our optimizations. sybilccFBA is a package in R that provides an improved implementation of MOMENT (MetabOlic Modeling with ENzyme kineTics). We used version 3.0.1. Lingo (https://www.lindo.com/index.php/products/lingo-and-optimization-modeling) is a commercial optimization software. Lingo version 13 was used as the alternative solver to double check the results from BARON.

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

## Acknowledgements

We thank Uri Alon, Matteo Mori, and Terence Hwa for helpful discussions and Abdelmoneim Desouki for providing the sybilccFBA R package. This work was supported by the Volkswagen Foundation under the "Life?" initiative, and by the German Research Foundation (DFG) through grants IRTG 1525, CRC 680, CRC 1310, and, under Germany's Excellence Strategy, through grant EXC 2048/1 (Project ID: 390686111).

## Author contributions

H.D. conceived of the study. X.P.H. developed, implemented, and parameterized the model and performed the analyses. P.S. prepared Fig. 1 and the S.B.M.L. model description. M.J.L. supervised the study. X.P.H. and M.J.L. interpreted the results and wrote the manuscript.

## Funding

## Competing interests

The authors declare that they have no competing interests.
