## [Peer Review File · Nature Communications]

Reviewers' comments:

Reviewer #1 (Remarks to the Author):

In their manuscript "The protein translation machinery is expressed for maximal efficiency in Escherichia coli", Hu and colleagues describe an ab-initio, optimality-based computational model of protein translation in E. coli. The model describes resource allocation to ribosomes, elongation factors, and tRNAs based on the assumption that the relevant limiting resource is total mass concentration. All parameter values stem from the literature (or were obtained independently from measurable data). Mathematically, the model is formulated as a non-linear optimality problem, which was solved numerically. In the paper, the authors discuss a number of model predictions regarding (growth rate dependent) abundances of translation machinery components. Results for cells under chloramphenicol stress are shown in the supplement.

The manuscript is very well written. It clearly points out the modelling assumptions, data provenance, the way the model was built, and the context of other existing modelling approaches, and the predictions are remarkably accurate.

What I like in particular is the way the optimality problem handles - implicitly - resource allocation between translation and other cell processes and their common effect on growth. In contrast to the "bacterial growth law" models, the authors predefine a given growth rate and determine the minimal amount of translation machinery that is needed to achieve this growth rate, implying that all other cell processes (including metabolism) need to run on the remaining resources. Turning this around, one can argue: whenever cells do achieve a certain growth rate in reality, they must have found a way to satisfy this metabolic requirement, and thus a (potentially optimal) compromise between translation and metabolism. This yields a valid, and surprisingly simple description of the translation/metabolism trade-off and of the resulting possible growth rates.

I have no major criticisms regarding the paper.

Minor remarks:

(1) The authors provide the model in the form of an excel table and of an executable GAMS file, but I could not find a file with the actual model results (predicted translation machinery abundances). Since readers may not have access to GAMS, and since the calculations seem to be heavy (8 hours runtime), I suggest to add another table with the numerical results (machinery abundances) for all figures shown in the paper, along with the experimental / experiment-derived values shown in the Figures, for comparison.

(2) I also suggest that the authors provide their model at BioModels database in a MIRIAM-compliant SBML format. For a well-structured model like this (even though it is large), this should be doable - but if the authors strongly oppose this idea, I will not insist on this point.

(3) Line 80 "At this required protein production rate, we minimized the combined cost of the translation machinery in our model, treating the concentrations of all components as free variables;" -> at this point, it would be good to mention that this is a nonlinear optimality problem that needs to be solved numerically.

(4) For me, the previous point also raises the question: is it possible that this problem turns out to be convex (given that the amino acid incorporation rates, and therefore many fluxes in the model, are predefined)? In the optimisations, were there any indications of local optima? If so, how does the algorithm deal with this?

(5) The most striking deviations between predictions and experimental data appear in Figure 3 and supplementary Figure 3 at the largest growth rates. Are there obvious explanations for these

deviations, or is it just SOMETHING that's not right about model details or optimality assumption?

(6) Figure 1 is not very clear. In the model description (SI), the ribosome acts as an "enzyme". In the graphics, however, it looks like it is "consumed and produced" (whereas mRNA, which also formally acts as an "enzyme", is depicted very differently). In the plot, there should be a clear meaning to the arrows. Some arrows (supposedly describing a single reaction) look like they referred to several, separate reactions. In case of doubt, the authors should consider using the SBGN conventions.

(7) Generally, I think that Fig 1 is missing a lot of the model details (also how different codons and amino acids are treated). The authors should redo this Figure from scratch (maybe with subfigures), trying to give a really good overview of all the ideas and details behind the model.

(8) The term "codon label" in Figure 1 is not clear to me. Is this a biological notion, or just a notion used to formulate the model?

(9) SI "and glucose was chosen as the reference to" -> "and growth on glucose .."?

(10) SI "we fitted the mRNA concentration as a quadratic function of growth rate" -> I guess the authors mean a second-order polynomial, not a pure quadratic function? Please clarify

(11) SI Line 102: "The concentration of amino acids encoded by codon-i" - I don't fully understand the logic: if two codons code for one amino acid, then one would get two formulae, and thus conflicting values for the concentration of the amino acid, right?

(12) SI Line 113 "assuming that the mass fraction of protein in total dry weight was 50%" - In Scott & Hwa 2010, the ratio "total RNA / total protein" varies between 0.1 and 0.5, depending on the growth rate - to me it seems unlikely, that at the same time, the protein mass fraction remains constant at 50%.

(13) SI Line 152: In the step from free to active ribosome, I don't see why a MM kinetics should be used - since it's a simple binding reaction, I would rather expect a bi-molecular mass-action kinetics which, together with the termination reaction, might yield a MM-like formula for the fraction of active ribosomes. Especially, the authors state that the mRNA concentration is much greater than the free ribosome concentration, which is not the typical case for a usage of MM kinetics.

(14) SI Line 187: "distinguished by codon labels:" Please describe what these codon labels are (also in the main text)

(15) SI Line 193 "In the model, there are 61 codons and 40 tRNAs," This is not clear at all in Figure 1

(16) SI Line 200 "translocated to the next round of elongation" -> "Translocated" is not fully clear to me

(17) SI Line 207: "ribosome that binding" typo?

(18) SI Line 375: "mRNA and tRNA were assayed at different growth rates;" -> So the mRNA and tRNA data stem from other growth conditions, but with comparable growth rates? Please clarify

Wolfram Liebermeister

Reviewer #2 (Remarks to the Author):

The paper by Hu et al. describes a model for understanding the levels of expression of components of the translation machinery, their growth rate dependence and the underlying evolutionary optimization. Hu et al. propose that these can be understood by considering a minimization of the costs of the machinery under the constraint of a given protein synthesis rate.

I enjoyed reading this paper, it is well-written and stimulates thoughts. Nevertheless I am not completely convinced by the authors' claims and, specifically, whether their approach differs from previous ones in terms of the biological content or only in terms of formulation of the model. Below are a few points, all somewhat interdependent, which in my opinion require clarification.

1) A clear shortcoming of the current approach (which is however shared with others) is the missing link to metabolism. The model uses protein concentration at a given growth rate as a constraint, in reality however, the protein concentrations, the composition of the translation machinery and the growth rate are all output of the dynamics of the expression system and dependent on the nutrient concentrations in the growth medium.

Thus, the current model is less microscopic and mechanistic than suggested by the authors - it is mechanistic with respect to the translation machinery, but uses growth rate and protein (or amino acids) concentration as constraints rather than as mechanistic inputs or (even better) outputs of the model.

2) The authors claim that mass density of a cell is the main cost parameter. While I understand that compared to other models that considered protein synthesis as the main cost, this allows them to associate a cost with RNA, which accounts for a substantial fraction of the translation apparatus, this approach seems to imply that the synthesis rate itself is not a cost factor. For long-lived components, this should not make much difference in the output of the model, however, the underlying interpretation of the economic principles are very different. This becomes most clear in the extreme case of a high turnover rate that results in the same concentration. If the main cost is in the synthesis, the turnover rate would make a big difference, if cost is in the mass density, turnover would be irrelevant. There is also experimental evidence for a cost to synthesis: Stoebel et al. *Genetics* 178, 1653 (2008).

3) Related to the previous two points, I am wondering whether the cost minimization under constraints can be thought of as a principle of the cell, subject to evolution, or whether this should rather be thought of as an optimization principle for the modeler who wants to obtain a description of the cell. So to what extent does the agreement between the current model and experimental data tell us something about the optimization principles in place in the cell?

4) Again related to that, how different is the constrained minimization of cost from an unconstrained maximization of growth rate under a given growth medium (e.g. Scott et al. *Mol. Sys. Biol.* 2014)? I assume the two approaches are very different based on their microscopic description, but maybe less so at the level of growth-rate dependencies: for growth rate dependence, one has to vary a parameter that influences growth (a nutrient parameter in the model of Scott et al and one of the constraints in the current model) and determine the quantity of interest as well as the growth rate both as functions of that parameter, but plot them against each other. Should any differences be expected or are these just two different mathematical ways of looking at the same thing?

Reviewer #1 (Wolfram Liebermeister)

1. In their manuscript "The protein translation machinery is expressed for maximal efficiency in *Escherichia coli*", Hu and colleagues describe an ab-initio, optimality-based computational model of protein translation in *E. coli*. The model describes resource allocation to ribosomes, elongation factors, and tRNAs based on the assumption that the relevant limiting resource is total mass concentration. All parameter values stem from the literature (or were obtained independently from measurable data). Mathematically, the model is formulated as a non-linear optimality problem, which was solved numerically. In the paper, the authors discuss a number of model predictions regarding (growth rate dependent) abundances of translation machinery components. Results for cells under chloramphenicol stress are shown in the supplement.

The manuscript is very well written. It clearly points out the modelling assumptions, data provenance, the way the model was built, and the context of other existing modelling approaches, and the predictions are remarkably accurate.

What I like in particular is the way the optimality problem handles - implicitly - resource allocation between translation and other cell processes and their common effect on growth. In contrast to the "bacterial growth law" models, the authors predefine a given growth rate and determine the minimal amount of translation machinery that is needed to achieve this growth rate, implying that all other cell processes (including metabolism) need to run on the remaining resources. Turning this around, one can argue: whenever cells do achieve a certain growth rate in reality, they must have found a way to satisfy this metabolic requirement, and thus a (potentially optimal) compromise between translation and metabolism. This yields a valid, and surprisingly simple description of the translation/metabolism trade-off and of the resulting possible growth rates.

I have no major criticisms regarding the paper.

Response: We thank the reviewer for the positive evaluation.

Minor remarks:

(1) The authors provide the model in the form of an excel table and of an executable GAMS file, but I could not find a file with the actual model results (predicted translation machinery abundances). Since readers may not have access to GAMS, and since the calculations seem to be heavy (8 hours runtime), I suggest to add another table with the numerical results (machinery abundances) for all figures shown in the paper, along with the experimental / experiment-derived values shown in the Figures, for comparison.

Response: A local optimum is typically found already after ~1min. We allow the solver to search for a total of 8h to find a global optimum; however, in no simulations was the first local optimum replaced, indicating that the optimization problem may possibly be convex.

Action: We now explain the relevance of the 8h search limit (SI line 440), and we added the table as suggested (Supplementary Table 3).

(2) *I also suggest that the authors provide their model at BioModels database in a MIRIAM-compliant SBML format. For a well-structured model like this (even though it is large), this should be doable - but if the authors strongly oppose this idea, I will not insist on this point.*

Response: We agree that it is desirable to provide the model in a re-usable, structured SBML form.

Action: We constructed an SBML file that describes all molecular species, parameters, and reactions in the model and uploaded it to the Biomodels database (id: MODEL2006210001). It has the following specifications:

- (1) SBML Level 3 version 1.
- (2) Species, reactions, and parameters are included.
- (3) Species annotation: Each species was annotated by (i) the SBO (system biology ontology) term and (ii) its ID in a public database or a publication (KEGG ID for metabolites, UniProt ID for proteins, Ecocyc ID for ribosome, and publications for tRNA and ternary complex).
- (4) Reaction annotation: The reactions and parameters were annotated by (i) the SBO term and (ii) the references (the detailed reactions included in our model are not listed in any public reaction database).
- (5) Costs. The costs (molecular weight only) are described as parameters and annotated with SBO terms.
- (6) The fixed translation rate (input of each optimization), the steady state assumption (flux balance), and the objective function (total mass concentration) are not included in the SBML file, but have to be provided by the modeler.

The content of the SBML file is summarized in the SI (SI line 17).

(3) *Line 80 "At this required protein production rate, we minimized the combined cost of the translation machinery in our model, treating the concentrations of all components as free variables;" -> at this point, it would be good to mention that this is a nonlinear optimality problem that needs to be solved numerically.*

Action: done (line 101).

(4) *For me, the previous point also raises the question: is it possible that this problem turns out to be convex (given that the amino acid incorporation rates, and therefore many fluxes in the model, are predefined)? In the optimisations, were there any indications of local optima? If so, how does the algorithm deal with this?*

Response: We ran all simulations for 8 hours to identify a global optimum; however, in no case did the solver identify multiple local optima (including when we tried alternative solvers). Thus, it appears likely that the problem is indeed convex, though we did not show this formally. *That the problem may be convex appears reasonable, as it is similar to the problem examined by Noor et al. (The Protein Cost of Metabolic Fluxes: Prediction from Enzymatic Rate Laws and Cost Minimization, PLOS Comp Biol 12: e1005167).*

Action: We now discuss the possibility that the problem is convex briefly in the manuscript (line 102), and in slightly more detail in the Supplementary Information (SI line 440).

(5) *The most striking deviations between predictions and experimental data appear in Figure 3 and supplementary Figure 3 at the largest growth rates. Are there obvious explanations for these deviations, or is it just SOMETHING that's not right about model details or optimality assumption?*

Response: Based on Hugo Dourado's analysis of optimal balanced growth (Nature Communications 1226 (2020)), it is *conceivable* that it is the increased dilution of intermediates at high growth rates that causes these discrepancies; however, we were not able to test this directly.

Action: We now discuss this possibility in the text (line 145).

(6) *Figure 1 is not very clear. In the model description (SI), the ribosome acts as an "enzyme". In the graphics, however, it looks like it is "consumed and produced" (whereas mRNA, which also formally acts as an "enzyme", is depicted very differently). In the plot, there should be a clear meaning to the arrows. Some arrows (supposedly describing a single reaction) look like they referred to several, separate reactions. In case of doubt, the authors should consider using the SBGN conventions.*

Response: We agree that the previous version of Fig. 1 was not as consistent and as clear as it could have been.

Action: We redrew the modeled reaction network in Fig. 1 as suggested, using the SBGN conventions.

(7) *Generally, I think that Fig 1 is missing a lot of the model details (also how different codons and amino acids are treated). The authors should redo this Figure from scratch (maybe with subfigures), trying to give a really good overview of all the ideas and details behind the model.*

Action: We redrew Fig. 1 from scratch as suggested, providing a graphical model representation that is both more systematic and more complete.

(8) *The term "codon label" in Figure 1 is not clear to me. Is this a biological notion, or just a notion used to formulate the model?*

Response: "Codon label" is a term we introduced to describe the model. It refers to the specific codon (from the mRNA being translated) that is presented by the ribosome at a given moment. However, we agree that this term might be misleading.

Action: We avoided the term "codon label" in the new Fig. 1, and we now always refer to the "codon presented by the active ribosome" in the text (e.g., SI line 326).

(9) *SI "and glucose was chosen as the reference to" -> "and growth on glucose .."?*

Response: This was indeed a typo.

Action: Corrected.

(10) SI "we fitted the mRNA concentration as a quadratic function of growth rate" -> I guess the authors mean a second-order polynomial, not a pure quadratic function? Please clarify.

Response: Indeed – we are sorry for the incorrect statement.

Action: Corrected (SI line 66, line 91, line 136, line 574).

(11) SI Line 102: "The concentration of amino acids encoded by codon- i " - I don't fully understand the logic: if two codons code for one amino acid, then one would get two formulae, and thus conflicting values for the concentration of the amino acid, right?

Response: The value defined in SI Eq. (3) is not the total concentration of an amino acid. If codon- i codes for amino acid AA_j , then $c_{\text{codon-}i}$ is the concentration of AA_j in cellular proteins that were encoded by codon- i in the corresponding mRNA. The total concentration of any amino acid is given by the sum of the $c_{\text{codon-}i}$ values for all codons encoding this amino acid.

Action: This is now clarified in the text (SI line 120, line 414).

(12) SI Line 113 "assuming that the mass fraction of protein in total dry weight was 50%" - In Scott & Hwa 2010, the ratio "total RNA / total protein" varies between 0.1 and 0.5, depending on the growth rate - to me it seems unlikely, that at the same time, the protein mass fraction remains constant at 50%.

Response: Indeed, experiments show that the protein fraction in dry weight varies between ~75% and ~50% from low to high growth rates (Ref. 6,27).

Action: For the revised manuscript, we repeated all simulations, accounting for the variable protein concentration by an appropriate scaling with the growth rate (see subsection starting on SI line 40, SI line 130).

(13) SI Line 152: In the step from free to active ribosome, I don't see why a MM kinetics should be used - since it's a simple binding reaction, I would rather expect a bi-molecular mass-action kinetics which, together with the termination reaction, might yield a MM-like formula for the fraction of active ribosomes. Especially, the authors state that the mRNA concentration is much greater than the free ribosome concentration, which is not the typical case for a usage of MM kinetics.

Response: Our original (handwaving) explanation of the origin of the MM kinetics for the ribosome activation through binding to mRNA was indeed not correct. The detailed derivation of the rate law can be found in Borkowski et al. (Ref. 29 in SI; see its Appendix 2 and Table 2). Briefly, the model comprises two steps for initiation and one step for elongation and found that at steady state, the rate of protein translation initiation follows Michaelis–Menten kinetics with the free ribosome in the “substrate” position and mRNA in the “enzyme” position of the rate law. The k_{cat} (K_{1i} in Borkowski et al.) and K_m (K_{2i} in Borkowski et al.) are combinations of multiple parameters of the underlying elementary reactions that are assumed to follow mass-action kinetics.

Action: We corrected the discussion of the MM kinetics for ribosome activation (SI line 292).

(14) SI Line 187: "distinguished by codon labels:" Please describe what these codon labels are (also in the main text)

Response: “Codon label” is a term we introduced to describe the model (see response to comment (8)). It refers to the specific codon (from the mRNA being translated) that is presented by the ribosome at a given moment. However, we agree that this term was not very intuitive.

Action: We replaced “codon label” by *expressions such as* “codon presented by the ribosome” throughout the text.

(15) SI Line 193 *"In the model, there are 61 codons and 40 tRNAs," This is not clear at all in Figure 1.*

Response: We have to agree (again) that Fig. 1 was not a good representation of the model.

Action: We redrew Fig. 1 from scratch. We now explicitly show a representative set of codon / tRNA combinations; the full set would make the figure too crowded. In the figure caption as well as in the text (line 86) we stress that the figure shows only a subset of the total 61 codons and 40 tRNAs.

(16) SI Line 200 *"translocated to the next round of elongation" -> "Translocated" is not fully clear to me*

Action: We converted the statement to “Simultaneously, the Ribosome_{codon-i} is converted to Ribosome_{active}, which is ready to participate in the next round of elongation” (SI line 341).

(17) SI Line 207: *"ribosome that binding" typo?*

Response: Yes.

Action: Corrected to “ $c_{\text{ribo-codon-}i}$ is the concentration of ribosomes that present codon- i (Ribosome_{codon-i})” (SI line 349)

(18) SI Line 375: *"mRNA and tRNA were assayed at different growth rates;" -> So the mRNA and tRNA data stem from other growth conditions, but with comparable growth rates? Please clarify*

Response: Each panel in Supplementary Figure 1 shows experimentally estimated vs. predicted concentrations of the different components of the translation machinery (ribosome, EF-Tu, EF-Ts, tRNA, mRNA) in one growth condition. Because ribosome, EF-Tu, and EF-Ts concentrations were assayed in the same proteomics experiment (Schmidt *et al.*), and because we to use the proteome composition (assumed to be predominantly a function of growth rate) as a boundary condition for each optimization, we chose the conditions examined in the proteomics study as the basis for the individual panels this figure. mRNA and tRNA were assayed in conditions with growth rates that differ from those of the proteomics experiment. To be able to still plot mRNA and tRNA data in the same panels, we fitted second order polynomial regression models to the available data for mRNA and tRNA concentrations, respectively, and then used the regressions to estimate the concentrations at the growth rates corresponding to the panels in Supplementary Figure 1. Absolute mRNA concentration was only assayed for growth rates between 0.11/h and 0.49/h, and we did not attempt to extrapolate mRNA concentrations beyond this range.

Action: We improved the explanation of this procedure in the legend of Supplementary Figure 1 (SI line 572).

Reviewer #2

The paper by Hu et al. describes a model for understanding the levels of expression of components of the translation machinery, their growth rate dependence and the underlying evolutionary optimization. Hu et al. propose that these can be understood by considering a minimization of the costs of the machinery under the constraint of a given protein synthesis rate.

I enjoyed reading this paper, it is well-written and stimulates thoughts. Nevertheless I am not completely convinced by the authors' claims and, specifically, whether their approach differs from previous ones in terms of the biological content or only in terms of formulation of the model. Below are a few points, all somewhat interdependent, which in my opinion require clarification.

- 1) A clear shortcoming of the current approach (which is however shared with others) is the missing link to metabolism. The model uses protein concentration at a given growth rate as a constraint, in reality however, the protein concentrations, the composition of the translation machinery and the growth rate are all output of the dynamics of the expression system and dependent on the nutrient concentrations in the growth medium.*

Thus, the current model is less microscopic and mechanistic than suggested by the authors - it is mechanistic with respect to the translation machinery, but uses growth rate and protein (or amino acids) concentration as constraints rather than as mechanistic inputs or (even better) outputs or the model.

Response: We agree that it would be desirable to find the optimal states of a genome-scale mechanistic model that combines metabolism and protein translation (and possibly other central cellular processes, such as transcription and DNA replication). Here, we restrict the mechanistic analysis exclusively to the components of the translation machinery, for two reasons. First, while excellent stoichiometric models of *E. coli* metabolism exist, the kinetic parameters for a majority of the metabolic reactions are currently unknown. Thus, a genome-scale model as envisaged by the Reviewer would have to be parameterized with largely arbitrary (or, possibly worse, fitted) kinetic parameters; it could thus not be truly mechanistic. In contrast, a very detailed and fully parameterized mechanistic model for translation can be assembled from the literature.

Second, translation is a complex, central process at the heart of biological cells. We have a good understanding of the workings of its machinery's individual components, and a large body of quantitative experimental data on their abundances in different conditions is available. But the translation apparatus is a "machine" with many degrees of freedom, where the same rate of protein production could be achieved with very different relative abundances of its components. It is still unclear according to which organizing principle(s) – if any – these abundances are set by the cell. This is what our work aims to explore.

Action: We now make this motivation and the limitations of our approach more explicit in the introduction of the revised manuscript (line 27, line 48).

2) *The authors claim that mass density of a cell is the main cost parameter. While I understand that compared to other models that considered protein synthesis as the main cost, this allows them to associate a cost with RNA, which accounts for a substantial fraction of the translation apparatus, this approach seems to imply that the synthesis rate itself is not a cost factor. For long-lived components, this should not make much difference in the output of the model, however, the underlying interpretation of the economic principles are very different. This becomes most clear in the extreme case of a high turnover rate that results in the same concentration. If the main cost is in the synthesis, the turnover rate would make a big difference, if cost is in the mass density, turnover would be irrelevant. There is also experimental evidence for a cost to synthesis: Stoebel et al. Genetics 178, 1653 (2008).*

Response: In our previous submission, we only considered costs that are proportional to the masses of molecules. This choice of a cost function was motivated by the consideration of molecular crowding and the very good agreement with experimental data both for individual enzymes and their substrates (Ref. 19) and in the submitted manuscript. However, the biological fitness of *E. coli* depends on many factors, and hence any simple assignment of fitness costs to the cellular investment into molecules can only be approximate. As we had not tested other potential cost functions, it was not clear if molecular masses are the only choice that resulted in such good agreement. While the work of Dourado *et al.* (Ref. 19) indicates that costs of enzymes and their products are proportional to their molecular masses, the analysis of Stoebel *et al.* indicates that costs lie in the production rather than the abundance of proteins. We thus agree with the Reviewer that it is desirable to examine a set of alternative cost functions.

Action: This we have done in the revised manuscript: in addition to molecular masses, we now examine molecular costs that are proportional to

- the protein mass fractions (as examined by Terry Hwa's group, Refs. 5,9,10,33);
- the carbon content (Ref. 21);
- the amount of ATP required for the production of proteins and functional RNA (Refs. 22,23);
- the amount of catalysts (protein or RNA) required for the production of proteins and functional RNA (Ref. 24).

All components whose concentrations we predict consist of protein, RNA, or both, and all costs examined are approximately proportional to the lengths of RNA and protein molecules. Thus, the relative costs of all components are essentially a function of the RNA/protein cost ratio r , i.e., the cost of RNA per nucleotide divided by the cost of protein per amino acid. For tRNA and rRNA, this cost ratio is broadly similar between molecular masses ($r=3.0$), carbon content ($r=2.0$), ATP cost ($r=1.6$), and synthesis costs ($r=1.7-2.1$) across minimal growth conditions ($\mu < 1 \text{ h}^{-1}$; **Supplementary Fig. 7**). In contrast, assuming that costs are proportional to only the protein content of the molecular assemblies^{9,20} results in an RNA/protein cost ratio of zero. Predictions based on protein costs hence overestimate mRNA and tRNA concentrations (which cost nothing), with corresponding underestimates of EF-Ts and especially EF-Tu concentrations (**Supplementary Figures 5, 6**).

We conclude that regardless of the exact cost function that influences *E. coli* fitness, the cost of the translation machinery appears to be (near) optimal. These new findings are reported in the manuscript (line 76, line 103, line 174), including the abstract (line 20). Details are given in the SI (starting line 161).

3) *Related to the previous two points, I am wondering whether the cost minimization under constraints can be thought of as a principle of the cell, subject to evolution, or whether this should rather be thought of as an optimization principle for the modeler who wants to*

obtain a description of the cell. So to what extent does the agreement between the current model and experimental data tell us something about the optimization principles in place in the cell?

Response: In the model, we minimize the combined costs of the translation machinery's components, resulting in quantitative agreement between predictions and experimental data – in the absence of adjustable parameters. The translation machinery accounts for a substantial amount of the dry mass of *E. coli* cells at larger growth rates, and it is hence likely that natural selection has shaped the abundances of its components. Accordingly, the most parsimonious explanation of our result is that natural selection has minimized a cost function that is similar to the cost functions examined by us (mass, ATP investment, or macromolecular investment into production). However, all we can rigorously say is that our results are consistent with such an evolutionary optimization. This hypothesis could be tested, e.g., through lab evolution experiments under strong selection for growth rate in a constant environment.

Action: We briefly discuss these issues in the revised manuscript (line 214).

4) *Again related to that, how different is the constrained minimization of cost from an unconstrained maximization of growth rate under a given growth medium (e.g. Scott et al. Mol. Sys. Biol. 2014)? I assume the two approaches are very different based on their microscopic description, but maybe less so at the level of growth-rate dependencies: for growth rate dependence, one has to vary a parameter that influences growth (a nutrient parameter in the model of Scott et al and one of the constraints in the current model) and determine the quantity of interest as well as the growth rate both as functions of that parameter, but plot them against each other. Should any differences be expected or are these just two different mathematical ways of looking at the same thing?*

Response: Scott *et al.* (2014) use a phenomenological model based on the „bacterial growth laws” most prominently described in Scott *et al.* (Science 2010). This model considers two proteome sectors, a ribosomal sector with proteome fraction Φ_R and a metabolic sector with proteome fraction Φ_P , responsible for the production of the amino acids consumed by the ribosomal sector. A constraint relates these two proteome sectors to the maximally available proteome fraction for protein synthesis, Φ_R^{\max} , which is assumed to be constant: $\Phi_R + \Phi_P = \Phi_R^{\max}$. Thus, the model of Scott *et al.* has only one free parameter, the proteome fraction allocated to the ribosomal sector, Φ_R . For a given Φ_R , the growth rate (which is defined by the rate of protein production) is set by three phenomenological parameters: the “translational efficiency” γ , the “nutritional efficiency” ν , and a constant proteome fraction of inactive ribosomes, Φ_R^{\min} . Maximizing the growth rate under these constraints results in an optimal ribosomal proteome fraction.

The approach of Scott *et al.* is thus very different from the one presented in our manuscript in the modelled details: 2 phenomenological, coarse-grained “reactions” with linear kinetics for Scott *et al.*, compared to of 274 characterized biochemical reactions with experimentally characterized kinetics in our manuscript.

Both approaches are interested in answering broadly the same question: given that resources are limited (proteome fractions for Scott *et al.*, catalyst and reactant masses etc. in our manuscript), what is the optimal way to distribute cellular resources in order to allow fast growth? Scott *et al.* approach this question by maximizing the growth rate while assuming (i) a constant, known translational efficiency γ (translation rate per ribosomal proteome fraction) and (ii) an environment-dependent constant, the nutritional efficiency (amino acid production rate per metabolic proteome fraction). In reality, the translational efficiency is unlikely to be constant across growth rates, and both parameters are not derived from a mechanistic

description, but are derived from fits to coarse-grained experimental data. What Scott *et al.* are able to show with this approach is that under relatively simple assumptions, an optimal allocation of proteome mass to translation and metabolism exists, and the relationship between the ribosomal proteome fraction and the growth rate is qualitatively similar to that observed experimentally, i.e., is linear.

Thus, while the model of Scott *et al.* is powerful when the parameters are fitted to experimental data, it is unclear what its mechanistic basis is. In particular, there is no clear explanation for the existence (and the size) of the “offset” of the ribosomal proteome fraction at zero growth rate in this model, Φ_R^{\min} .

While we minimize the cost of translation rather than maximizing growth rate, our approach is mathematically equivalent to a maximization of growth rate under a constraint on the total cost and under certain additional assumptions (such as a constant amino acid composition of the proteome across growth rates). Both approaches vary some condition-dependent parameters (the nutritional efficiency for Scott *et al.*, the proteome mass and composition in our manuscript) and then optimize an aspect of cellular resource allocation. However, in contrast to Scott *et al.*, we are not interested in the relative global resource allocation between translation and biosynthesis, but in the quantitative pattern of resource allocation across different components of the translation machinery.

In sum, beyond being more detailed and aiming at predicting the abundance of translation machinery components rather than the overall, relative investment into translation and biosynthesis, our model differs from that of Scott *et al.* in the following characteristics: (i) its equations and parameters have clear biochemical and biophysical interpretations; and (ii) we make quantitative predictions for observable concentrations and reaction rates without any adjustable parameters.

Action: We have extended the discussion of the work of Klumpp *et al.* in Supplementary Note 1 (SI line 459) by now also discussing the relationship between our work and the model of Scott *et al.*, which forms the basis for Klumpp *et al.*'s model; we refer to this discussion on line 44.

REVIEWERS' COMMENTS:

Reviewer #1 (Remarks to the Author):

The authors have fully and satisfactorily addressed all my previous comments.

I appreciate the time and effort they put into providing a high-quality SBGN version of Figure 1, and congratulate them to the high-quality, well-annotated SBML model they deposited at BioModels.net.

Wolfram Liebermeister

Reviewer #2 (Remarks to the Author):

I already stated in my first report that this is an intriguing, thought-stimulating paper. The questions raised in the report were convincingly addressed by the reviewers, I recommend to accept the paper-

Response to Reviewers

Reviewer #1 (Wolfram Liebermeister):

The authors have fully and satisfactorily addressed all my previous comments.

I appreciate the time and effort they put into providing a high-quality SBGN version of Figure 1, and congratulate them to the high-quality, well-annotated SBML model they deposited at BioModels.net.

Response: We thank the Reviewer for this positive assessment.

Reviewer #2:

I already stated in my first report that this is an intriguing, thought-stimulating paper. The questions raised in the report were convincingly addressed by the reviewers, I recommend to accept the paper-

Response: We thank the Reviewer for this positive assessment.